# Superlim:
# A Swedish Language Understanding Evaluation Benchmark

**Aleksandrs Berdicevskis**[*]  **Gerlof Bouma**[*]  **Robin Kurtz**[†]  **Felix Morger**[*]  **Joey Öhman**[‡]

{aleksandrs.berdicevskis,gerlof.bouma,felix.morger}@gu.se
robin.kurtz@kb.se  joey.ohman@embark-studios.com

**Yvonne Adesam**[*]  **Lars Borin**[*]  **Dana Dannélls**[*]  **Markus Forsberg**[*]  **Tim Isbister**[§]
**Anna Lindahl**[*]  **Martin Malmsten**[†]  **Faton Rekathati**[†]  **Magnus Sahlgren**[§]  **Elena Volodina**[*]

**Love Börjeson**[†]  **Simon Hengchen**[‖]  **Nina Tahmasebi**[¶]

[*] Språkbanken Text, University of Gothenburg
[†] KBLab, National Library of Sweden   [‡] Embark Studios
[§] AI Sweden   [‖] iguanodon.ai   [¶] University of Gothenburg

## Abstract

We present Superlim, a multi-task NLP benchmark and analysis platform for evaluating Swedish language models, a counterpart to the English-language (Super)GLUE suite. We describe the dataset, the tasks, the leaderboard and report the baseline results yielded by a reference implementation. The tested models do not approach ceiling performance on any of the tasks, which suggests that Superlim is truly difficult, a desirable quality for a benchmark. We address methodological challenges, such as mitigating the Anglocentric bias when creating datasets for a less-resourced language; choosing the most appropriate measures; documenting the datasets and making the leaderboard convenient and transparent. We also highlight other potential usages of the dataset, such as, for instance, the evaluation of cross-lingual transfer learning.

## 1 Introduction

In recent years, the move towards Transformer-based large pretrained language models has brought the need for comprehensive and standardized multi-task benchmarking in NLP into focus. Since these models, through their generic pre-training setup, can be fine-tuned on many different tasks, a test suite which covers a range of different NLP domains is required to measure overall performance. For this reason, several different benchmarks have been proposed. For English, the most notable suites arguably are the (Super)GLUE benchmark (Wang et al., 2018, 2019) for natural language understanding and GEM (Gehrmann et al., 2021) for natural language generation.

In this paper, we present Superlim, a multi-task benchmark for Swedish natural language understanding. It contains 14 different datasets which cover 15 tasks ranging from word understanding tasks, such as word analogy and word similarity, to text classification tasks, such as natural language inference and sentence acceptability. The release of this benchmark comes at a time when multiple pre-trained language models have been created for Swedish, and larger models are under construction (ranging from Malmsten et al., 2020 to Ekgren et al., 2023). In this active research environment, a multi-task benchmark is crucial to gauge state-of-the-art performance and guide future development. For Swedish this is especially important, since limitations (financial, computational and related to data availability) are more considerable than for English.

Creating a benchmark for Swedish comes with its own unique set of practical challenges and methodological considerations. While there exist high-quality Swedish datasets that can be reused, not all of them have been created with the task of evaluating large language models in mind. Here, the relatively close linguistic relationship to English comes with possibilities of translating (or adapting) existing datasets. Such approaches do however potentially introduce linguistic and cultural Anglocentric bias into Swedish language benchmarking. Therefore, Superlim combines datasets of three kinds: a) preexisting ones, reformatted and revised, b) translated or adapted from English, and c) created from scratch specifically for Superlim.

To facilitate progress tracking and sharing of results, models and datasets, we create an online leaderboard for Superlim, which will serve as a platform for submitting and publishing results, and

release reference implementations with baseline results for a collection of existing Swedish language models.

The rest of the paper looks as follows: In Section 2 we briefly cover related work. Section 3 describes the tasks included in Superlim and the corresponding datasets, and Section 4 their standardized documentation. Section 5 motivates Superlim's main evaluation measure: Krippendorff's $\alpha$. The online design of the leaderboard is the subject of Section 6. Sections 7 and 8 describe the reference implementations and the baseline results. Section 9 provides details on where to find Superlim and Section 10 concludes the paper.

## 2 Related work

The already mentioned (Super)GLUE has inspired the creation of similar benchmarks for other languages: CLUE (Xu et al., 2020) and CLiMP (Xiang et al., 2021) for Chinese, Russian SuperGLUE (Shavrina et al., 2020), KLEJ (Rybak et al., 2020) and LEPISZCZE (Augustyniak et al., 2022) for Polish, ParsiNLU for Persian (Khashabi et al., 2021), IndoNLU for Indonesian (Wilie et al., 2020) and NorBench (Samuel et al., 2023) for Norwegian. Several suggestions to move to other evaluation approaches have been voiced (Ethayarajh and Jurafsky, 2020; Ribeiro et al., 2020; Kiela et al., 2021), but the GLUE-like benchmarks still remain the default approach.

Superlim is not the only benchmark for Swedish. Very recently, ScandEval has been released (Nielsen, 2023). This multilingual suite contains four tasks for Swedish. Some of the datasets used for the evaluation contain synthetic data (for instance, incorrect sentences created by scrambling word order in the correct sentences), which is not necessarily problematic, but presumably less valuable than natural data. Another benchmark for Swedish, OverLim (Kurtz, 2022), is entirely machine translated from English.

## 3 Tasks and datasets

We have three types of tasks: text level (Section 3.1), word level (Section 3.2) and diagnostic (Section 3.3). Describing the creation of all datasets in detail is beyond the scope of this paper. Instead we focus on briefly describing the datasets, their tasks and any substantial changes that have been made to the original sources for the purposes of Superlim. For more detailed informa-

tion about the datasets and their creation, we refer to the documentation that accompanies Superlim, itself described in Section 4. Table 1 lists all the tasks in Superlim.

Most datasets are divided into predefined training, development and test splits. All *test sets* are of gold-standard quality: they have been manually annotated from scratch or manually corrected after automatic processing. Some of the *training* and *development sets* may contain non-corrected data, for instance automatically translated data. Our rationale here is that a model can be trained on anything, as long as it yields high, robust, unbiased performance. Evaluation, however, must always occur on gold data to serve its purpose. An important caveat, however, is that those test sets that were automatically translated and then manually corrected may suffer from so-called post-editese (Toral, 2019). In all cases, the editing was carefully done by native speakers with background in NLP and/or linguistics, who were aware of this potential problem. We hope that these factors have at least mitigated it.

The word-level datasets were mostly created with non-contextualized, type level approaches in mind (for instance, static word vectors like those obtained from word2vec). These therefore do not contain large training splits, but small training sets that can be used to calibrate otherwise unsupervised methods, or for few-shot training generative models. The diagnostic datasets only have test splits. Here, the training and development splits of SweNLI are to be used.

### 3.1 Text-level tasks

#### 3.1.1 Absabank-Imm

Absabank-Imm is derived from the Swedish aspect-based sentiment (stance) analysis corpus described in Rouces et al. (2020). We selected the part annotated for author attitude regarding immigration in Sweden, quantified as real-valued scores between 1 (very negative) and 5 (very positive). These scores are the arithmetic mean of integer scores from individual annotators, but since most paragraphs were labelled by only one annotator, most average values are also integer.

The data for this task comes from different sources: editorials and opinion pieces from the national Swedish newspapers *Svenska dagbladet* and *Aftonbladet*, and forum posts from *Flashback*, a large Swedish discussion forum.

| Task | Name | Type | Data source(s) | Translated | dataset size (items) | | |
|---|---|---|---|---|---|---|---|
| | | | | | **train** | **dev** | **test** |
| **Text level** | | | | | | | |
| stance | Absabank-Imm | score | Newspapers and forum | no | 3 898 | 487 | 487 |
| argumentation | Argument. sents | 3-lab | Common Crawl | yes | 3 450 | 750 | 1 065 |
| acceptability | DaLAJ-GED | 2-lab | Learner essays, textbooks | no | 35 581 | 4 702 | 4 371 |
| similarity | SweParaphrase | score | STS Benchmark | yes | 5 715 | 1 499 | 1 378 |
| summarization | SweDN | gen | Newspaper | yes | 29 847 | 4 529 | 3 745 |
| QA | SweFAQ | select | Websites of authorities | no | 781 | 110 | 109 |
| entailment | SweNLI | 3-lab | Various sources | yes | 392 702 | 9 815 | 305 |
| word meaning | SweWiC | 2-lab | SALDO, Eukalyptus, Wiktionary | no | 4 486 | 500 | 1 000 |
| coreference | SweWinograd | 2-lab | Winograd | yes | 721 | 135 | 140 |
| **Word level** | | | | | | | |
| relatedness | SuperSim rel'ness | score | SimLex-999, WordSim353 | partly | 131 | – | 1 229 |
| similarity | SuperSim similarity | score | SimLex-999, WordSim353 | partly | 131 | – | 1 229 |
| analogy | Swedish Analogy | select | English Analogy, Wiktionary | partly | 2 045 | – | 18 593 |
| synonymy | SweSAT | select | Scholastic Aptitude Test | no | 83 | – | 739 |
| **Diagnostics** | | | | | | | |
| entailment | SweDiagnostics | 3-lab | News, Wikipedia, ACL, Reddit | yes | – | – | 1 104 |
| entailment | SweWinogender | 3-lab | Winogender | yes | – | – | 624 |

Table 1: Overview of the tasks in Superlim. Task type is one of: binary labelling, ternary labelling, scoring (regression), answer selection (multiple choice, pairing) or text generation. A dataset counts as translated when the training and test items were translated (manually or automatically) from its English counterpart.

### 3.1.2 Argumentation sentences

In Argumentation sentences, the task is to categorize sentences into argument *for*, argument *against*, or *unrelated* to a given topic. The topics include abortion, death penalty, nuclear power, marijuana legalization, minimum wage and cloning. The data for this task was machine translated from the argument unit recognition and classification dataset described in Trautmann et al. (2020) and manually corrected by a native speaker of Swedish.

### 3.1.3 DaLAJ-GED

For DaLAJ-GED, sentences have to be classified as being grammatically correct or not. Incorrect datapoints were collected from the Swedish learner corpus SwELL (Volodina et al., 2019). Sentences containing errors from this material were manipulated to contain only one error per datapoint – a sentence with multiple mistakes can thus give rise to multiple datapoints. Correct datapoints were sampled from the same material and from the course book corpus COCTAILL (Volodina et al., 2014).

The DaLAJ-GED materials contain additional annotation, such as the type and location of an error, its recommended correction, and information about the sentence source. This additional annotation is not currently used in the task itself, but we consider it valuable for error analysis, and it is a potential basis for future tasks using this material.

### 3.1.4 SweParaphrase

The SweParaphrase dataset consists of sentence pairs, and the task is to estimate their semantic similarity, that is, to what extent they express the same state of affairs on a scale from 0 (dissimilar) to 5 (similar). The gold standard annotation is an arithmetic mean of multiple judgements.

The dataset is based on the Semantic Textual Similarity Benchmark (STS-B) (Cer et al., 2017), which was first automatically translated to Swedish using the Google Translate API (Isbister et al., 2021) and then manually corrected by a native speaker. The similarity values were taken from STS-B without any adjustment.

### 3.1.5 SweDN

SweDN (Monsen and Jönsson, 2021) is a summarization task, the only sequence generation task included in Superlim. This dataset collects almost 40 thousand articles from the Swedish newspaper *Dagens Nyheter*. The lead paragraph in each article serves as its ground truth summary.

### 3.1.6 SweFAQ

The dataset for SweFAQ consists of answers to frequently asked questions taken from websites of nine Swedish authorities, such as the Social Insurance Agency and the Swedish Tax Agency. The dataset contains 976 question-answer pairs that fall into 100 different categories, for instance *COVID-*

*19 vaccination* or *parental benefits*. Each datapoint in the dataset is a question and a list of all answers from the same category in a randomized order. The task is to select the matching answer, by giving its index. The length of the list of question-answer pairs varies across categories.

### 3.1.7 SweNLI

SweNLI is a natural language inference task: for two sentences, a premise and a hypothesis, predict the relation between the two (neutral, contradiction or entailment). The SweNLI dataset is derived from two existing resources. The test split was constructed from a grammar-based automatic Swedish translation of the FraCaS test suite (Cooper et al., 1996; Ljunglöf and Siverbo, 2012). For Superlim, the Swedish FraCaS suite was extensively manually revised and culturally adapted to Swedish real-world facts.

For the training and development splits, a machine translated version of the Multi-genre Natural Language Inference (Williams et al., 2018) dataset is used. The original data was collected from ten different texts ranging from transcribed telephone calls to magazine articles. We translated the dataset to Swedish using the OPUS-MT machine translation framework (Tiedemann and Thottingal, 2020).

### 3.1.8 SweWiC

SweWiC is modelled after the word-in-context task described in Pilehvar and Camacho-Collados (2019). A polysemous target word is provided in two contexts. The system's task is to determine whether the word is used with the same meaning or not.

The development and test splits use example sentences collected for the Swedish lexical resource SALDO (Borin et al., 2013), and sentences taken from the Eukalyptus corpus (Adesam et al., 2015), which has word-sense annotation based on SALDO. Training data was constructed from Swedish Wiktionary.[1] Since the word-in-context task doesn't directly rely on the sense inventory itself, but only asks whether two instances use the same meaning or not, differences between Wiktionary and SALDO in terms of sense inventory are acceptable (see also Pilehvar and Camacho-Collados, 2019).

### 3.1.9 SweWinograd

SweWinograd is a coreference resolution task, cast as a binary mention-pair classification problem.

---

[1]See https://sv.wiktionary.org/.

Each datapoint contains a fragment in which a pronoun and a possible antecedent are highlighted. The system has to predict whether they are coreferent or not.

SweWinograd was created by manual translation of the Winograd schema challenge (Levesque et al., 2012), as included in SuperGLUE. Note that the Winograd schema challenge is also present in the original GLUE benchmark, but as an entailment task.

## 3.2 Word-level tasks

For each of the word-level tasks described below, we created small training splits by randomly selecting 10% of the combined data, meant to enable few shot learning or calibration of existing models.

### 3.2.1 SuperSim relatedness and SuperSim similarity

Superlim includes two tasks based on the the SuperSim dataset (Hengchen and Tahmasebi, 2021). SuperSim contains word pairs annotated for semantic *similarity* and for semantic *relatedness*, both with scores between 0 (low degree) and 10 (high degree). Semantic similarity is the extent to which two concepts share semantic properties, semantic relatedness refers to a more general association between the concepts. For instance, *cup* and *coffee* are related, but not similar. SuperSim relatedness and SuperSim similarity concern scoring pairs for relatedness and similarity, respectively.

SuperSim's word pairs were translated from SimLex-999 (Hill et al., 2015) and WordSim-353 (Finkelstein et al., 2002) and then labelled by five native speakers of Swedish. The ground truth is the arithmetic mean of the labels, but the individual labels are preserved, too.

### 3.2.2 Swedish Analogy

The Swedish Analogy dataset (Adewumi et al., 2022) consists of analogies of the form "*Stockholm* is to *Sverige* ('Sweden') as *Berlin* is to *Tyskland* ('Germany')." The dataset contains different categories of analogy, of semantic (real world facts, as above, or lexical semantic relations) or of morphological nature. The dataset is partly based on Mikolov et al. (2013) for English, but was enriched with additional categories and translated to Swedish using automatic tools and Wiktionary matches, after which it was proof-read by native speakers.

In Swedish Analogy, analogies have to be completed by predicting the fourth element given the first three. The set of items from which the model may select an answer is not limited by the task, but only by the vocabulary of the model itself.

### 3.2.3 SweSAT

SweSAT was created in the context of Superlim, and collected from the synonyms part of the Swedish Scholastic Aptitude Test (*Högskoleprovet*) for the years 2006–2021. The task is, given a word or an expression, to select the best synonym from a list of five candidates.

### 3.3 Diagnostic tasks

Superlim comes with two diagnostic tasks, which are both cast as inference tasks. Participating systems should use the training materials supplied for SweNLI.

### 3.3.1 SweDiagnostics

SweDiagnostics is a is a manually translated version of the GLUE diagnostic dataset (Wang et al., 2018). The dataset is used for diagnosing a system's ability to handle specific linguistic phenomena. The diagnostic task is cast as an inference task, where the premise and hypothesis are identical but for a minimal manipulation, like the insertion of negation. The entailment relationship (entailment, neutral, or contradiction) relies on the linguistic phenomenon targeted. There are 33 fine-grained phenomena across four coarse-grained categories (lexical semantics, predicate-argument structure, logic and common sense). Translating the English dataset, we made sure that the targeted linguistic phenomena are present even in the resulting Swedish sentences and that the sentences are idiomatic Swedish.

In SuperGLUE, the same diagnostic dataset was used, but formulated as a binary classification problem. In Superlim, we have chosen to keep the original ternary version.

### 3.4 SweWinogender

SweWinogender is an inference task designed to diagnose gender bias in models. It is a reformulation of the pronoun resolution test set of Hansson et al. (2021), which, in turn, was manually translated from/inspired by Winogender (Rudinger et al., 2018). We mirror SuperGLUE in this choice, since it has an inference task reformulation of the English Winogender.

Items in SweWinogender consist of a premise, a short text containing a pronoun that allows only one reasonable interpretation of this pronoun, and an hypothesis that spells out some interpretation of this pronoun. For correctly resolved pronouns, there is an entailment relation between the two sentences. Each premise-hypothesis pair occurs three times in the dataset, each time with a different pronoun: *han* 'he', *hon* 'she', or *hen* '(singular) they'. For a model that does not display any gender bias, its performance on the inference task should not correlate with the choice of pronoun.

## 4 Documentation

The importance of documenting datasets cannot be overestimated. Still this step is often overlooked in the NLP field. Since some parts of Superlim are revised versions of pre-existing datasets, it is particularly important to be explicit about what changes were made. We therefore devised a *documentation sheet*: a template inspired by Gebru et al.'s (2021) "Datasheets" and Pushkarna et al.'s (2022) "Data cards". Our template, however, is more compact than the former and is specifically adapted to the Superlim context. Documentation sheets consist of six sections:

1. Identifying information: basic information about the dataset and its creators;
2. Usage: why the dataset was created and how it can be used;
3. Data: the description of the contents (including basic statistics, format, data source, collection method, inter-annotator agreement etc.);
4. Ethics and caveats: ethical considerations, potential pitfalls, discouraged usage;
5. About documentation: information about the documentation and dataset versions.
6. Other information, including a bibliography.

Each Superlim task is accompanied by a documentation sheet. The compactness of the template is intentional: it improves the chances that documentation will actually be written, maintained and read. An example can be found in Appendix A.

## 5 Evaluation measures

A benchmark containing a mix of task types will almost by necessity involve a wide range of evaluation methods and measures. The appropriate choices depend on the tasks themselves and which

insights we would like to get into our models. Good research practice tells us to evaluate models in those ways that teach us most about them. However, in a benchmark "competition" setting, were we would like to summarize a model's performance with one final score, this leads to a problem. In GLUE, this is dealt with by providing an average over very different beasts like accuracy, Matthews correlation coefficient, f-score, and Spearman's correlation coefficient (that have different ranges and different intepretations). In the context of the Superlim benchmark, we explore an alternative: the use (where possible) of a single family of measures, Krippendorff's $\alpha$ (Krippendorff, 2018).

Krippendorff's $\alpha$ is a measure of inter-annotator agreement, for instance used in work on linguistic annotation (Paun et al., 2022). We use $\alpha$ as an evaluation measure by treating the system to be evaluated as one annotator, and the gold-standard creators as another. A perfect system always agrees with the gold-standard creators ($\alpha = 1$), whereas a poor system shows no systematic agreement with them ($\alpha \approx 0$). Even lower scores ($-1 \leq \alpha < 0$) are signs of systematic mistakes.

Krippendorff's $\alpha$ is parameterized by a distance metric (Krippendorff, 2011) which allows us to use it on different types of task. For binary and ternary labelling tasks, we use nominal-$\alpha$. In the binary case, nominal-$\alpha$ linearly maps to macro-averaged f-score. For scoring tasks we use interval-$\alpha$. Since it is a measure of *agreement*, interval-$\alpha$ is stricter than correlation measures like Pearson's $r$, which do not require a system to get the sizes of predicted scores correct, only the shape of their distribution. For a benchmark, we find this strictness a desirable property. The selection tasks do not allow for a direct application of $\alpha$. However, we can calculate a derived measure, a pseudo-$\alpha$, by pretending, after the fact, that the selection tasks are binary decision tasks and by calculating a nominal-$\alpha$ for this recast task. We present a more elaborate investigation of $\alpha$ as an evaluation measure in a separate paper (Bouma, 2023).

We currently do not have an $\alpha$-based measure for the summarization task SweDN, for which the benchmark uses ROUGE-1.

SweWinogender is evaluated using nominal-$\alpha$, but in addition also using *gender parity*, which measures how consistently the model answers across datapoints that only differ in the choice of pronoun.

## 6 Leaderboard

The benchmarking results are collected at the Superlim leaderboard, where both development and test set results are made available. In addition to the evaluation outcomes on the different tasks, the leaderboard also lists information about a participating model's type and size (in number of parameters), and the size of the dataset used to pre-train the model.

The Superlim project strives for transparency and openness, not just with regards to our datasets but also to the contents of the leaderboard. As part of this, we link participating models' predictions on development and test sets as well as scripts and configuration details used to produce those results. This facilitates replication of results and confirmation of their validity. There is an additional point with this level of transparency: The gold labels of the Superlim test sets are not hidden. This makes evaluation easier, but it also makes cheating (intentional or accidental, see discussion in the limitations section) very easy. We hope that asking participants to be open about their methods can act as a countermeasure to this potential problem.

The design of the leaderboard is also guided by the idea that there is no single best model (Ethayarajh and Jurafsky, 2020). The choice of a model for some application sometimes need only depend on the performance on certain subsets of our evaluation suite and can even be restricted to only included models up to a certain size. We therefore allow users of the leaderboard to filter and sort out all the tasks and models that are irrelevant for their specific needs: users can include and exclude tasks, filter out model families, sort by size, overall performance, and task-specific performance.

Consider the following use case: a developer wants to build a proofreading application using a transformer model, and needs their application to run as fast as possible on limited hardware. The most relevant task in Superlim for them is DaLAJ-GED, and the developer may wish to restrict themselves to models with less than 150M parameters. In the leaderboard, they can now easily select all *base* models, exclude all word-level tasks to declutter the table, and sort according to performance on DaLAJ-GED. That would bring on top the `KBLab/bert-base-swedish-cased-new` model, that otherwise performs worse than average. The developer further checks if the performance on the validation set matches the performance on

| Model | Type | Size | Language | Model | Avg std |
|---|---|---|---|---|---|
| KB/bert-base-swedish-cased | BERT | base | sv | AI-Nordics/bert-l-sw-c | 0.013 |
| KBLab/bert-base-swedish-cased-new | BERT | base | sv | KBLab/mt-bert-l-sw-c-165k | 0.018 |
| KBLab/megatron-bert-base-swedish-cased-600k | BERT | base | sv | KB/bert-b-sw-c | 0.018 |
| KBLab/megatron-bert-large-swedish-cased-165k | BERT | large | sv | KBLab/mt-bert-b-sw-c-600k | 0.034 |
| AI-Nordics/bert-large-swedish-cased | BERT | large | sv | NbAiLab/nb-bert-base | 0.056 |
| xlm-roberta-base | RoBERTa | base | multi | xlm-roberta-base | 0.085 |
| xlm-roberta-large | RoBERTa | large | multi | KBLab/bert-b-sw-c-new | 0.122 |
| AI-Sweden/gpt-sw3-. . . | GPT | 126M–40B | sv, no, da, is, en | xlm-roberta-large | 0.324 |

Table 2: Language models used for the reference implementation

Table 3: Average standard deviation of performance across hyper-parameter configurations

the test set, before finally choosing a small set of suitable models to evaluate on their new application's data. A screenshot of the leaderboard taken from this scenario can be found in Figure 1 in Appendix B.

# 7 Reference implementation

To create baselines for Superlim, we built a number of reference implementations using the Swedish language models listed in Table 2, all hosted at HuggingFace. In addition to these models, we also provide results for a set of non-neural supervised machine learning methods, including SVM, Decision Trees, and Random Forests. Random choice and majority label give lower bounds.

We used the following hyperparameters when fine tuning the language models, along with the default arguments in the HuggingFace Trainer class:[2] warm-up ratio 0.06, weight decay 0.1 (0.0 for GPT models), number of training epochs 10, fp16. We use early stopping with patience 5 for the number of epochs. Learning rate and batch size were tuned for all tasks and models, from the following search space: learning rate [1e−5,2e−5,3e−5,4e−5], batch size [16,32]. Because of the size of the training set, we only considered the lowest and highest learning rate for SweNLI. The hyperparameter search gives us a performance score (Krippendorff's $\alpha$) for each task, model and configuration of hyperparameters. The average standard deviation of these scores are summarized in Table 3, which shows xlm-roberta-large to be by far most sensitive to these settings.

We created reference implementations for all text-level tasks except for SweDN.

[2]huggingface.co/docs/transformers/main_classes/trainer

Static word vectors[3] were used as an approach to SuperSim relatedness and SuperSim similarity by scaling the word pairs' cosine similarities to 0–10. SweSAT was addressed by picking the choice with the highest cosine similarity. For Swedish Analogy we first applied vector algebra to calculate a vector for the fourth (missing) item, and then used cosine similarity to pick the closest vocabulary item.

The two word-level selection tasks were also solved using the gpt-sw3-... models (Ekgren et al., 2023). Swedish Analogy by inspecting log-probabilities of a Swedish translation of the text *a synonym is being sought for the search term. The term* x *serves as a synonym for* y, where *x* is a choice item and *y* the question's target expression, and choosing the most probable item. SweSAT employed a few-shot learning strategy to prompt the GPT model to complete the analogy.

# 8 Results and discussion

The results for the word-level tasks are in Table 4. The difference between SuperSim relatedness and SuperSim similarity for the static word vectors mirrors the difference in the random baseline for these tasks, but it contrasts with results when measured in terms of correlation (Hengchen and Tahmasebi, 2021), which are much more even. Here we see the sensitivity of $\alpha$ to the size of the predicted scores: cosine similarity systematically overestimates semantic similarity in SuperSim similarity. The GPT models show a clear effect of model size on Swedish Analogy and SweSAT, and the largest models do well on these tasks. The smallest GPT model is however not more effective than static word vectors on SweSAT.

Table 5 gives the results for the text-level tasks. The different tasks by and large "agree" with each

[3]https://spacy.io/models/sv/#sv_core_news_lg

| Model | Word-level task | | | |
|---|---|---|---|---|
| | SuperSim rel | SuperSim sim | Swedish Analogy | SweSAT |
| AI-Sweden/gpt-sw3-126m | | | 0.400 | 0.340 |
| AI-Sweden/gpt-sw3-1.3b | | | 0.732 | 0.665 |
| AI-Sweden/gpt-sw3-20b | | | 0.824 | 0.820 |
| AI-Sweden/gpt-sw3-40b | | | **0.838** | **0.854** |
| Random | -0.037 | -0.302 | 0.000 | 0.000 |
| Static word vectors | **0.410** | **-0.109** | 0.013 | 0.365 |
| **Task type** | score | score | select | select |

Table 4: Reference model performance for the word-level tasks, reported as Krippendorff's $\alpha$.

| Model | Text-level task | | | | | | | | **Avg** | Diagnostics |
|---|---|---|---|---|---|---|---|---|---|---|
| | Absabank | Arg Sent | DaLAJ | Paraphrase | FAQ | NLI | WiC | Winograd | | |
| KBLab/mt-bert-l-sw-c-165k | 0.508 | **0.628** | **0.753** | 0.874 | **0.777** | 0.231 | 0.308 | 0.189 | **0.534** | 0.393 |
| AI-Nordics/bert-l-sw-c | 0.480 | 0.563 | 0.745 | 0.862 | 0.719 | **0.241** | 0.316 | **0.192** | 0.515 | 0.347 |
| KB/bert-b-sw-c | **0.529** | 0.555 | 0.740 | 0.845 | 0.641 | 0.179 | **0.376** | 0.139 | 0.501 | 0.349 |
| xlm-roberta-large | 0.516 | 0.584 | 0.738 | **0.882** | 0.584 | 0.205 | 0.367 | 0.081 | 0.494 | **0.415** |
| KBLab/mt-bert-b-sw-c-600k | 0.449 | 0.562 | 0.718 | 0.867 | 0.709 | 0.218 | 0.277 | 0.061 | 0.483 | 0.363 |
| NbAiLab/nb-bert-base | 0.390 | 0.541 | 0.644 | 0.823 | 0.660 | 0.172 | 0.326 | 0.120 | 0.459 | 0.314 |
| KBLab/bert-b-sw-c-new | 0.428 | 0.554 | **0.753** | 0.755 | 0.447 | 0.163 | 0.140 | 0.042 | 0.410 | 0.338 |
| xlm-roberta-base | 0.366 | 0.497 | 0.701 | 0.813 | 0.473 | 0.186 | 0.181 | -0.177 | 0.380 | 0.318 |
| SVM | 0.286 | 0.354 | 0.518 | 0.239 | 0.038 | 0.000 | 0.042 | 0.055 | 0.192 | 0.026 |
| Decision Tree | 0.117 | 0.156 | 0.269 | 0.200 | 0.040 | 0.192 | 0.040 | -0.240 | 0.074 | 0.037 |
| Random | 0.008 | 0.013 | 0.007 | -0.043 | -0.150 | -0.091 | -0.010 | 0.081 | -0.038 | 0.004 |
| Random Forest | 0.005 | -0.272 | -0.312 | 0.143 | 0.032 | -0.411 | 0.003 | -0.177 | -0.124 | 0.010 |
| Majority label/Avg | -0.052 | -0.272 | -0.340 | -0.001 | -0.310 | -0.434 | -0.333 | -0.177 | -0.240 | -0.404 |
| **Task type** | score | 3-lab | 2-lab | score | select | 3-lab | 2-lab | 2-lab | | 3-lab |

Table 5: Reference model performance for the text-level tasks, sorted by average Krippendorff's $\alpha$, and overall diagnostic score. See the main text for discussion of the SweWinogender results, here omitted for space reasons.

other in their ranking of the different models: models scoring high on one task are likely to score high on another. We also see that the models characterized as large on average outrank the other models. The scores are relatively low overall. We take this not just as an artefact of our choice of evaluation measure, which in most cases is stricter than, say, accuracy or f-score, but also as an indication that Superlim is a difficult benchmark. We agree with Bowman and Dahl (2021) that this is a desirable property.

Particularly low scores are achieved for the inference task SweNLI and the pronoun resolution task SweWinograd. The bad performance for the former is connected to the low scores on both diagnostic tasks, which use the same training data and are of the same task type. Overall performance on SweDiagnostics maxes out at $\alpha$=.415, a breakdown of SweDiagnostics results can be found in Appendix C.

There is no column for the SweWinogender diagnostic task in Table 5: all models receive perfect gender parity scores but very low $\alpha$-s of around -0.3. This is the result of outputting the same label for basically all test items. As mentioned above, SweNLI and the diagnostic tasks are inference tasks that share the same, automatically translated training dataset. It is tempting to assume performance is affected by low quality of the training set. We note, however, that Argumentation sentences also is a ternary labelling task with automatically translated training data, and that the models fare much better there. It is likely that these are just hard tasks: also in Bowman and Dahl (2021) it is noted that GLUE's diagnostic set was challenging even for the best models, and SuperGLUE's co-reference task has the lowest baseline performance amongst the non-diagnostic tasks.

The models perform best on the paraphrase task SweParaphrase.

## 9 Distribution

All datasets constituting Superlim are available under Creative Commons licenses (CC BY 4.0,

CC BY-SA 4.0, respectively), and can be downloaded together with their documentation sheets from Språkbanken Text's website[4] or accessed through HuggingFace.[5] The Superlim leaderboard can be found at KBLab's website,[6] and includes model predictions as well as detailed information about the participating models.

## 10  Conclusion

We have presented Superlim, a Swedish benchmark and analysis platform for natural language understanding in the style of GLUE and SuperGLUE. Superlim contains 13 regular tasks and 2 diagnostic tasks from a broad range of task types, including word-level tasks that are not commonly seen in GLUE-style benchmarks. All tasks come with gold-standard test data, with training and development data – albeit not always of gold standard quality – and with detailed, standardized documentation.

We have chosen to make Superlim's leaderboard flexible and transparent, which we hope may help promote the dissemination of knowledge about the capacities of natural understanding models for Swedish and knowledge about their construction.

Finally, by having comparable tasks to the de facto standard SuperGLUE, Superlim not only contributes to the development of natural language understanding for Swedish, but also to research into transfer learning and multilingual models, and thus to a much wider part of the field of natural language processing.

## Author contributions

The authors are divided into three groups (Berdicevskis to Öhman, Adesam to Volodina, and Börjeson to Tahmasebi), depending on their contribution to the project and the paper. Within each group, the contributions are considered to be of equal weight, and the author names are listed alphabetically. The division of labour between the three partner institutions was as follows: dataset collection and/or construction – Språkbanken Text; reference implementation – AI Sweden; leaderboard development – KBLab.

## Acknowledgments

The *Superlim* project was supported by Sweden's Innovation Agency (Vinnova, grant nos 2020-02523 and 2021-04165) and by *Nationella språkbanken* – jointly funded by the Swedish Research Council (2018–2024, grant no. 2017-00626) and 10 partner institutions.

Joey Öhman contributed to Superlim while at AI Sweden. The contributions of Simon Hengchen and Nina Tahmasebi have in part been funded by *Towards Computational Lexical Semantic Change Detection* (Swedish Research Council, 2019–2022, grant no. 2018-01184) and *Change is Key!* (Riksbankens Jubileumsfond, grant no. M21-0021). RISE Research Institutes of Sweden was involved in the project at its early stage.

We would like to thank everybody who contributed to the creation of datasets or in any other way helped us with the project, in particular Francisca Hoyer, Yousuf Ali Mohammed, Tosin Adewumi, Julius Monsen, Arne Jönsson, Peter Ljunglöf and Jacobo Rouces.

## Limitations

Unlike some other benchmarks, Superlim has no hidden data, which may lead to data leaks. The models (pre)trained on the data crawled from the Internet may have seen the test data. While we require the submissions to be as transparent and thoroughly documented as possible, it is not always possible to control what exactly the very large training sets contain (and often it is not even known by those who trained the model). There is a risk that this problem will exacerbate as recent evidence have shown that larger language models (equal to or larger than 6B parameters), such as the GPT-J model (Wang and Komatsuzaki, 2021) and Chat-GPT,[7] are capable of memorizing data verbatim (Carlini et al., 2023).[8] In order to better understand how this affects Superlim, we encourage future work exploration of how much of the Superlim data has been memorized by large language models.

We do not currently have a human baseline to put the performance of models on Superlim into perspective.

Some of the training sets in the Superlim were automatically translated and have not been manually corrected. The translated test sets have all been

---

[4] https://spraakbanken.gu.se/resurser/superlim
[5] https://huggingface.co/datasets/sbx/superlim-2
[6] https://lab.kb.se/leaderboard/results

[7] https://openai.com/chatgpt
[8] https://hitz-zentroa.github.io/lm-contamination/blog/

thoroughly corrected, but, of course, they still may suffer to some extent from translationese (Gellerstam, 1986) or its exacerbated variant post-editese (Toral, 2019).

It can be argued that Superlim is too heterogeneous, containing, on the one hand more traditional natural language understanding tasks such as NLI, sentiment analysis, semantic similarity at the sentence level as well as word-level tasks, which are less often used to evaluate Transformer models, on the other (and one generation task in addition).

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

## A Example Superlim-style documentation sheet

I. IDENTIFYING INFORMATION

| | |
|---|---|
| Title* | SweWinogender |
| Subtitle | A Swedish diagnostic set for gender bias in natural language inference models |
| Created by* | Yvonne Adesam (yvonne.adesam@gu.se), Gerlof Bouma (gerlof.bouma@gu.se) |
| Publisher(s)* | Språkbanken Text |
| Link(s) / permanent identifier(s)* | https://spraakbanken.gu.se/en/resources/swewinogender |
| License(s)* | CC BY 4.0 |
| Abstract* | The SweWinogender test set is diagnostic dataset to measure gender bias in coreference resolution/textual entailment. It is modelled after the English Winogender benchmark, and is released with reference statistics on the distribution of men and women between occupations and the association between gender and occupation in modern corpus material. |
| Funded by* | Vinnova (dnr 2020-02523, dnr 2021-04165); Språkbanken Text |
| Cite as | [1] |
| Related datasets | Part of the Superlim collection (https://spraakbanken.gu.se/en/resources/superlim) |
| | See SweWinogender v1.0 for a formulation of this task as a pronoun resolution problem. |
| | Based upon/partially translated from Winogender Schemas [2] |

II. USAGE

| | |
|---|---|
| Key applications | Diagnose gender bias in natural language inference systems |
| Intended task(s)/usage(s) | (1) Indirectly the pronoun interpretation task cast as a natural language inference problem: decide whether a discourse fragment containing a pronoun entails a sentence with the pronoun replaced by a candidate antecedent. |
| | (2) Compare system predictions between pronoun types (masc/fem/gender-neutral) |
| | (3) Compare system predictions with auxiliary statistics on gender and occupation |
| Recommended evaluation measures | (1) Krippendorff's alpha on binary label |
| | (2) "Gender parity": The proportion of triples of items differing only by the type of pronoun, that receives identical labels. See [3]. |
| | (3) Correlation (Spearman's rho); plotting/visual inspection. See [2]. |
| Dataset function(s) | Diagnostics |
| Recommended split(s) | Test data only |

III. DATA

| | |
|---|---|
| Primary data* | Text |
| Language* | Swedish |
| Dataset in numbers* | 624 test items from 104 templates. 312 positive cases ('entailment') and 312 negative cases ('neutral'). |

| | |
|---|---|
| Nature of the content* | The test items are constructed from short discourse templates that contain two participants: one referred to by occupation, and one either by a role description. Furthermore, the templates contain a pronominal reference to one of these participants. The templates are constructed such that the interpretation of the pronoun follows from (common sense) reasoning. Each template gives rise to 6 test items: 3 possibilities depending on whether the feminine ("hon/henne/hennes"), masculine ("han/honom/hans") or gender-neutral pronoun ("hen/hens") is used, 2 possibilities depending on whether the hypothesis is entailed or not. A natural language inference model that is not sensitive to gender biases should therefore answer the same way for a triple of test items that only differs in which pronoun they contain. |
| | The test set is accompanied by an auxiliary dataset that contains two sets of statistics on the association between occupation and gender for the occupations mentioned in the test set. These statistics were extracted from a real-world database and from a corpus, respectively. The auxiliary data can be used to study gender-occupation biases in the system more directly. |
| Format* | Test items: JSON Lines, with 1 test item per line. Test items are given as a pair of sentences ('premise' and 'hypothesis') and a 'label' attribute that says whether the hypothesis is entailed by the premise ('entailment') or not ('neutral'). The metadata ('meta') contains identifying information about the sentence template that generated the test item, and a 'tuple-id' that can be used to calculate parity. |
| | Auxiliary data: TSV file with one occupation per row. Gives the following columns of information: 1) occupation; 2) % female practitioners according to SCB; (3)–(5) % occurences in female-associated contexts using small/medium/large collocate sets. See [1] for an explanation of the different corpus measures. |
| Data source(s)* | The test items are loose translations and/or inspired by the Winogender Schemes of [2]. |
| | The auxiliary data was collected by the first authors of [1], in the context of an MA course. The real-world statistics on gender and occupation were compiled on the basis of Statistics Sweden/SCB's open data (CC BY 4.0). Where occupations do not map 1-1 to SCB's categorization scheme, the supplied statistics are averages over several relevant categories. See [1] for details. The corpus-based statistics on gender-association of occupations where compiled from the Swedish Culturomics Gigaword Corpus [4]. |
| Data collection method(s)* | See [1] |
| Data selection and filtering* | See [1] |
| Data preprocessing* | See [1] |
| Data labeling* | Test items contain gold-standard coreference data by design. |
| Annotator characteristics | Test item compilation: 1 native speaker of Swedish with PhD in computational linguistics, 1 near-native speaker of Swedish with PhD in (corpus) inguistics. |

## IV. ETHICS AND CAVEATS

| | |
|---|---|
| Ethical considerations | The auxiliary data contains information about the distribution between women and men across occupations, and therefore contains data about subpopulations. The data does not contain reference to individuals – neither directly nor indirectly. |
| Things to watch out for | This is meant as a diagnostic, not as a target for training. |
| | The diagnostic only concerns occupation and gender, and this is only one of the many ways gender bias may be present in a coreference resolution model. In the words of [2]: "[a]s a diagnostic test of gender bias, we view the schemas as having high positive predictive value and low negative predictive value; that is, they may demonstrate the presence of gender bias in a system, but not prove its absence." |
| | Although the test items contain a threeway distinction in the pronouns used (han [masc], hon [fem], hen [neutral], the auxiliary data is restricted to a binary gender perspective. For task (3) above, it may however be interesting to compare system predictions for the gender-neutral pronoun ("hen") items with the auxiliary statistics to better understand how a system handles resolution of this pronoun. |

## V. ABOUT DOCUMENTATION

| | |
|---|---|
| Data last updated* | 20230125 v2.0 |
| Which changes have been made, compared to the previous version* | Reformulation as a natural language inference task. |
| Access to previous versions | Earlier versions available from website. |
| This document created* | 20210614; Gerlof Bouma (`gerlof.bouma@gu.se`) |
| This document last updated* | 20230208; Gerlof Bouma (`gerlof.bouma@gu.se`) |
| Where to look for further details | - |
| Documentation template version* | v1.1 |

## VI. OTHER

| | |
|---|---|
| Related projects | - |
| References | [1] Hansson, Mavromatakis, Adesam, Bouma and Dannélls (2021): The Swedish Winogender Dataset. In Proceedings of the 23rd Nordic Conference on Computational Linguistics (NoDaLiDa), pp452–459. `http://www.ep.liu.se/ecp/178/052/ecp2021178052.pdf`
[2] Rudinger, Naradowsky, Leonard and Van Durme (2018): Gender bias in coreference resolution. In Proceedings of the 2018 Conference of the North American Chapter of the Association for Computational Linguistics: Human Language Technologies, Volume 2 (Short Papers), pp8–14. `https://doi.org/10.18653/v1/N18-2002`
[3] Wang, Pruksachatkun, Nangia, Singh, Michael, Hill, Levy and Bowman (2019): SuperGLUE: A Stickier Benchmark for General-Purpose Language Understanding Systems. In Advances in Neural Information Processing Systems 32. `https://papers.nips.cc/paper/2019/file/4496bf24afe7fab6f046bf4923da8de6-Paper.pdf` |

[4] Rødven Eide, Tahmasebi and Borin (2016): The Swedish Culturomics Gigaword corpus: A one billion word Swedish reference dataset for NLP. In Digital Humanities 2016. From Digitization to Knowledge: Resources and Methods for Semantic Processing of Digital Works/Texts, pp8–12. https://ep.liu.se/ecp/126/002/ecp16126002.pdf

# B   Leaderboard

Figure 1: A screenshot of the website showing the leaderboard, filtered to show only large language models of the 'base' category, sorted by performance on the DaLAJ-GED task on the validation (dev) set, with datasets for which the results are not available excluded.

# C   SweDiagnostics: detailed results

| Model | All | Coarse-grained categories | | | | Fine-grained categories | | | | | |
|---|---|---|---|---|---|---|---|---|---|---|---|
| | | LS | PAS | L | K | UQnt | MNeg | 2Neg | Coref | Restr | Down |
| xlm-roberta-large | **0.415** | **0.441** | **0.434** | **0.345** | 0.290 | 0.509 | **0.648** | 0.641 | **0.391** | 0.167 | -0.541 |
| KBLab/mt-bert-l-sw-c-165k | 0.393 | 0.368 | 0.430 | 0.305 | **0.314** | **0.669** | 0.434 | 0.641 | 0.334 | 0.115 | -0.541 |
| KBLab/mt-bert-b-sw-c-600k | 0.363 | 0.303 | 0.392 | 0.306 | 0.283 | **0.669** | 0.43 | 0.641 | 0.365 | 0.000 | -0.621 |
| KB/bert-b-sw-c | 0.349 | 0.319 | 0.354 | 0.282 | 0.273 | 0.582 | 0.366 | 0.433 | 0.349 | -0.114 | -0.348 |
| AI-Nordics/bert-l-sw-c | 0.347 | 0.310 | 0.348 | 0.308 | 0.239 | 0.510 | 0.646 | **0.799** | **0.391** | -0.114 | -0.548 |
| KBLab/bert-b-sw-c-new | 0.338 | 0.327 | 0.387 | 0.277 | 0.214 | 0.513 | 0.505 | 0.571 | 0.361 | -0.114 | -0.528 |
| xlm-roberta-base | 0.318 | 0.307 | 0.359 | 0.245 | 0.180 | 0.500 | 0.145 | 0.107 | 0.310 | -0.116 | -0.561 |
| NbAiLab/nb-bert-base | 0.314 | 0.305 | 0.358 | 0.270 | 0.144 | 0.582 | 0.578 | 0.339 | 0.262 | -0.057 | -0.463 |
| Decision tree | 0.037 | 0.000 | 0.041 | 0.047 | 0.028 | 0.555 | -0.247 | -0.141 | 0.113 | **0.266** | -0.169 |
| SVM | 0.026 | -0.006 | 0.000 | 0.047 | 0.003 | 0.319 | -0.366 | -0.429 | -0.032 | 0.080 | **-0.082** |
| Random forest | 0.010 | -0.034 | -0.052 | 0.006 | 0.067 | 0.347 | -0.284 | 0.328 | -0.099 | -0.127 | -0.169 |
| Random | 0.004 | -0.005 | 0.040 | 0.013 | 0.020 | 0.341 | 0.004 | -0.086 | 0.048 | -0.089 | -0.119 |
| Majority label/Avg | -0.404 | -0.378 | -0.482 | -0.376 | -0.350 | -0.300 | -0.351 | -0.626 | -0.411 | -0.600 | -0.579 |

Table 7: The results on the SweDiagnostics dataset. We report Krippendorf's $\alpha$ on the coarse-grained categories *Lexical Semantics* (LS), *Predicate-Argument Structure* (PAS), *Logic* (L) as well as *Knowledge* (K). In addition, five fine-grained categories, *Universal Quantifiers* (UQnt), *Morphological Negation* (MNeg), *Double Negation* (2Neg), *Anaphora/Coreference* (Coref), *Restrictivity* (Restr) and *Downward Monotone* (Down) are selected. The table reports categories identical to the ones reported in Table 5 in the GLUE paper (Wang et al., 2018), but note the difference in measure, which was $R_3$ in the cited paper.