# OpenReview forum: "Superlim: A Swedish Language Understanding Evaluation Benchmark"
_EMNLP/2023/Conference — EMNLP 2023 Main_

### Official Review · Reviewer_kbFp · 2023-08-04

**Soundness:** 5

**Excitement:**

4: Strong: This paper deepens the understanding of some phenomenon or lowers the barriers to an existing research direction.

**Paper Topic And Main Contributions:**

This paper presents a benchmark for the Swedish language models, inspired by GLUE and SuperGLUE. The authors detail the features of the benchmark, the collection and annotation of datasets used, and exemplify the use of such benchmark. This is a significant progress towards better Swedish models evaluation and development.

**Reasons To Accept:**

The paper is well written, presents a sound work, and makes very interesting comments on the methodology used to build the benchmark and the choice of the datasets, the limitation of using a single measure to compare models, and the anglocentric bias.

**Reasons To Reject:**

I don't see any reason to reject the paper.

**Reproducibility:**

4: Could mostly reproduce the results, but there may be some variation because of sample variance or minor variations in their interpretation of the protocol or method.

**Reviewer Confidence:**

4: Quite sure. I tried to check the important points carefully. It's unlikely, though conceivable, that I missed something that should affect my ratings.

**Typos Grammar Style And Presentation Improvements:**

- the unit of dataset sizes is not given in Table 1
- in stead -> instead
- ten different text -> ten different texts
- thedataset -> the dataset
- it performance -> its performance

---

> ### Author Rebuttal · Authors · 2023-08-28
>
> We thank the reviewer for their positive comments and will fix the raised grammar and presentation issues in a final version of the paper.

---

### Official Review · Reviewer_kEqu · 2023-08-05

**Soundness:** 4

**Excitement:**

3: Ambivalent: It has merits (e.g., it reports state-of-the-art results, the idea is nice), but there are key weaknesses (e.g., it describes incremental work), and it can significantly benefit from another round of revision. However, I won't object to accepting it if my co-reviewers champion it.

**Paper Topic And Main Contributions:**

This paper presents na new set of NLP tasks proposed as benchmark for evaluating general-purpose language models.

**Reasons To Accept:**

The range of selected tasks is sufficiently wide, it follows other benchmarking suites (e.g. GLUE) and overall seems to be well maintained -- I definitely support the "no hidden data" approach (NLP simply needs more evaluation data -- even if it is costly, not hiding parts of any existing ones as a lame excuse) as well as policy for the leaderboard as explained in the paper.

**Reasons To Reject:**

My only strong objections are related to the gold data creation and its reliability in terms of inter-annotator agreement. In the end the gold data is what everything depends on and it seem not to be described in very detail in the paper -- though I understand this is partially because of length limits. This concerns mainly the word-level tasks:
- anon-texttask-8, where sentences are taken from the Eukalyptus corpus (the 2016 paper on Eukalyptus corpus mentions only a fraction ["As of October, 2015, two annotators have annotated a total amount of 17,580 tokens, with an overlap of 2,919 tokens."] actually having more than one annotation. Has that changed/improved meanwhile? Even two annotations are a very poor standard, and I believe that by 2023 no single-person-annotated data should be part of any evaluation sets, particularly when it concerns tasks with a notoriously low inter-annotator agreement like WSD.
- anon-wordtask-1 and anon-wordtask-2, where you say "labelled by five native speakers of Swedish". What was their agreement/correlation?
- anon-wordtask-4 is missing any details how the SweSAT was used (how words were chosen, who annotated, what was the IAA -- if at all available)

Similarly, information about inter-annotator agreement (i.e. how many raters + raw numbers + maybe some chance-corrected coefficient) is completely missing from the documentation sheet (while I think it is the second most important quantitative information right after size :). Ideally all the tasks should be accompanied with information about human evaluation (where available).

Once the benchmark is here, many people will use it, and not many will look into it (unfortunately). I find it therefore important to raise these concerns now.

**Reproducibility:**

5: Could easily reproduce the results.

**Reviewer Confidence:**

5: Positive that my evaluation is correct. I read the paper very carefully and I am very familiar with related work.

**Typos Grammar Style And Presentation Improvements:**

- first paragraph in Introduction: "In recent years, the move towards Transformer based large pretrained language models has created a need for comprehensive and standardized benchmarking in NLP". Really? I do see you need to start the paper somehow, but starting it with a sentence that is basically false seems to be really awkward. The need has been there forever, not any smaller than these days!
- l.119 extra space in "In stead"
- l.418 the use OF? a...
- l.617 ABOUT? the participating models

---

> ### Author Rebuttal · Authors · 2023-08-28
>
> We thank the reviewer for their thoughtful comments. The reviewer raises concerns about the lack of information in the paper about dataset creation and reliability.
>
> Indeed, as the reviewer speculates, the descriptions of the tasks and their datasets are kept short because of the length of the paper. It is also related to the style of the paper, which is intended as a high level overview of the whole benchmark consisting of 13+2 tasks representing a range of NLU problems, and not as a detailed description of each task. We have therefore restricted details about the datasets to the documentation sheets. In addition, we note that some of the tasks already have their own associated publications with more detailed descriptions. This especially holds for included datasets that were created outside of the project.
>
> Having said that, we agree strongly with the reviewer that having detailed information about the annotation/construction process, and about the quality and reliability of the datasets is very important. The standardized documentation sheets specified for the benchmark were partly motivated by the desire to ensure such information is available and easy to locate.
>
> We will here answer the questions about the specific datasets raised by the reviewer, but reiterate that this information will be found in the documentation sheets. We will make sure to be clearer about this in the paper.
>
> * __anon-texttask-8, Eukalyptus corpus__ The Eukalyptus treebank as used for anon-texttask-8 has sense annotation for about ¾ of its ~100k tokens, each sense label accompanied by a quality indication. For anon-texttask-8, we only used the tokens labeled with the highest quality levels, where a) all of two or more annotators were unanimous, or b) the label had been vetted/corrected after a first annotation round by the leader of the annotation project.
> * __anon-wordtask-1 and anon-wordtask-2__ The data was taken as-is from the SuperSim dataset (Hengchen and Tahmasebi, 2021). This paper reports average pairwise Spearman’s ρ = 0.67 for similarity and ρ = 0.73 for relatedness.
> * __anon-wordtask-4__  The Swedish Scholastic Aptitude Test (Högskoleprovet) is a national test, held twice each year. We have taken one part of the Swedish SAT, the multiple choice synonym test, as the basis for our anon-word-task-4, from 15 years of past tests. These items, including the _by construction_ correct answers, are available from a website run by the government agency responsible for the SAT (the Swedish Council for Higher Education). No further selection/annotation was performed in the compilation of anon-wordtask-4.
>
> Regarding IAA information in the documentation sheets: there is no designated heading called "inter annotation agreement" or similar, but there _is_ room for this and related types of information in the documentation sheet, under the row headings "Data collection methods", "Data labeling", and "Annotator characteristics" (final rows in section III of the sheet), depending on the precise nature and context of the information.
>
> There is no mention of IAA in the example sheet in Appendix A, since IAA is not relevant for this dataset. It was not created by labelling/annotating given items, but rather by purposefully constructing and manipulating test items to have a preferred reading that either follows or is at odds with a gender-related bias. However other documentation sheets will include this information where relevant.
>
> In the final version of the paper, we will address the grammar and presentation issues raised by the reviewer. In particular we will fix the formulation of the the opening sentence to better communicate the intended meaning, which relates to the rise of LLMs that can be applied to a wide range of tasks and therefore call for benchmarks that deal with such a wide range.

---

### Official Review · Reviewer_4JdJ · 2023-08-11

**Soundness:** 5

**Excitement:**

4: Strong: This paper deepens the understanding of some phenomenon or lowers the barriers to an existing research direction.

**Missing References:**

Many of your datasets include machine-translated text that was post-edited by a human, and the introduction claims that they are gold-standard quality just like from-scratch annotations. While this is briefly addressed in the Limitations section (which I appreciate), I think you might be under-estimating the difference between post-edited MT and natural text. A citation to consider is:

Antonio Toral. 2019. Post-editese: an Exacerbated Translationese. In Proceedings of Machine Translation Summit XVII: Research Track, pages 273–281, Dublin, Ireland. European Association for Machine Translation.

**Paper Topic And Main Contributions:**

This paper presents a large multi-task benchmark for language understanding in Swedish, loosely similar to SuperGLUE, publicly released by the authors. The paper describes  the tasks and origin of the data, and the authors have developed a feature-rich leaderboard for tracking progress on the benchmark. The authors propose an evaluation measure that scores all tasks (with the exception of one) on the same scale. They also propose and utilize a new paradigm for documenting datasets, intended to incur lower overhead than existing paradigms. Finally, they provide reference implementations for most tasks and present baseline results.

**Questions For The Authors:**

Question A) Regarding line 238, could you provide more detail on how the data was "culturally adapted", in terms of who did it and what process they followed? I think that some of these details should be included in the text.

Question B) Regarding line 245, why was OPUS-MT used? I'm not directly familiar with the system but I suspect that its performance is subpar. This is especially confusing to me because the Google Translate API was used on another dataset.

Question C) For all datasets where something was translated or post-edited by one or more native speakers, can you provide more details on who those native speakers were? Were they authors of this paper, colleagues, professional translators, bilingual crowd-workers, or someone else? The resulting quality is likely to depend on this. This is especially important for the discussion at line 344.

Question D) What exactly are the values in Table 3? I gathered that the average is taken over various hyperparameter settings for a given model, but what distributions are you calculating standard deviations for? Are they, for example, distributions of mean benchmark scores for some number of randomized training runs?

Question E) Have you taken steps to mitigate the risk of accidental data leakage? I accept that preventing intentional cheating is out-of-scope, but from a practical standpoint it would be nice to know that accidentally training on test data is difficult. For example, putting the test inputs and their outputs in separate files might mitigate a web-crawl corpus from meaningfully leaking their correspondence.

**Reasons To Accept:**

 - NLP benchmarks in non-English languages are sorely needed.
 - The proposed benchmark's tasks provide broad coverage within the NLU domain.
 - The features included in the leaderboard and discussion behind the dataset documentation paradigm indicate that the authors thoughtfully considered the needs of multiple groups, including system developers and users.
 - The tasks and their source datasets are described with sufficient detail.

**Reasons To Reject:**

 - The potential negative impact of using post-edited test data may be underestimated. See the Missing References section.

**Reproducibility:**

5: Could easily reproduce the results.

**Reviewer Confidence:**

4: Quite sure. I tried to check the important points carefully. It's unlikely, though conceivable, that I missed something that should affect my ratings.

**Typos Grammar Style And Presentation Improvements:**

 - Line 13: I'm not sure what a "smaller language" is. I think readability would improve with a more precise phrase, such as "lower-resource language" or "less-supported language" (assuming I inferred your intent correctly).
 - Table 1: When I first looked at the bottom row, I didn't understand that I was looking at ditto marks (''). I later figured it out, and you don't necessarily need to change it, but I would recommend to double-check that your motivation is to improve clarity/readability instead of e.g. saving keystrokes or ink.
 - Line 35: "Anonymized Namea" -> "AnonymizedName, a"
 - Line 119: "In stead" -> "Instead"
 - Line 195: "extend" -> "extent"
 - Line 196: I think the idiom "state of affairs" hurts readability here. Consider just "meaning" or some other more direct phrase.
 - Table 3: The caption is not clear and is probably missing one or more words between "deviation" and "hyperparameter".
 - Lines 616-617: "information the" -> "information on the"
 - Line 645: "lead data" -> "lead to data"

---

> ### Author Rebuttal · Authors · 2023-08-28
>
> We thank the reviewer for their thoughtful comments. Below we will answer the direct questions posed by the reviewer. When it comes to the questions regarding specific datasets, we want to emphasize that this information will also be documented in the datasheets, which are part of the AnonymizedBenchmark project and will be linked to in the paper.
>
> __Question A, cultural adaptation of anon-texttask-7__ The data was culturally adapted by a native of speaker of Swedish, professor in NLP, by replacing English proper names (e.g. person names, toponyms, trademarks) and some other terms (e.g. job titles) with idiomatic Swedish equivalents.
>
> __Question B, choice for OPUS-MT__
> The processing of different datasets was carried out by different researchers at different points in time, and while there were universal project policies, some decisions were made independently. This is why it was possible for different machine translation systems to have been used. In this particular case, OPUS-MT was chosen because it is an open, transparent and accessible MT system, that gives reproducible results, which was taken as a more important characteristic than the possible improvement of translation quality that Google Translate potentially would offer.
>
> __Question C, translators/post-editors__ (For reasons of anonymity, we will not comment on whether dataset creators were co-authors, colleagues, etc.)
> * Anon-texttask-2 was post-edited by a native speaker of Swedish with a Master in language technology.
> * Anon-texttask-4 was post-edited by a native speaker of Swedish, a graduate student with a background in linguistics
> * Anon-texttask-7 was corrected and culturally adapted by a professor of language technology, who is a native speaker of Swedish with a PhD in linguistics.
> * Anon-texttask-9 was translated from the English / based upon English examples in committee by a native speaker and a near-native speaker  of Swedish with PhDs in (computational) linguistics.
> * Anon-diagnose-1 was post-edited by a native speaker of Swedish with a Master in linguistics.
> * Anon-diagnose-2 was translated from the English / based upon English examples in committee by a native speaker and a near-native speaker of Swedish with PhDs in (computational) linguistics.
>
> __Question D, Table 3__ For a given type of LLM and a given text-level task, there are 8 runs (2 batch sizes x 4 learning rates), each run gives a performance score as Krippendorff alpha. We then calculated the std dev over these runs. This tells us how much the performance of a single model on a single task varies with different hyperparameter settings, that is, it is an indication of how sensitive the model is to its hyperparameter settings on that task. The reported numbers in Table 3 give a summarized hyperparameter sensitivity score for each model by averaging over these task-specific std devs. We agree this is presented rather densely in the paper and will make an effort to improve the discussion of these results for the final paper.
>
> __Question E, data leakage__ Since many of the datasets are projects which have been released independently, we cannot guarantee that they have not already been collected by web crawlers. We will, however, take the reviewer's suggestion and split the labels into separate files on our end. Furthermore, we will encourage investigations into memorization of input and output data on the models using our benchmarks, in order to estimate the effect of accidental training on testing data. The following citation would be a good inspiration for this kind of work: Carlini, Nicholas, et al. "Quantifying memorization across neural language models." arXiv preprint arXiv:2202.07646 (2022). We will mention this in the final version of the paper.
>
> Finally, we share the reviewers concern that there can be a negative impact of using post-edited machine translated data, even though we do not know the degree of this impact. We have tried to lessen this impact on the benchmark by also including original Swedish datasets, and by conscious efforts of our translators/post-editors to make the translations sound more natural. Given the limited available annotated data in Swedish and the resources available for this project, it would not have been possible to include the current number and range of tasks without using post-edited data at all. The mentioned citation of Toral (2019) would make an excellent addition to the discussion in the Limitations section of our paper.
>
> In a final version, we will address the grammar and presentation issues that the reviewer raised.

---

### Meta-Review · Area_Chair_ZpUb · 2023-09-17

**Recommendation:** 5

**Metareview:**

This work presents a multi-task NLP benchmark and analysis platform for evaluating Swedish language models.

The three reviewers agree on the strong/excellent quality of the work with respect to soundness and excitement. The reviewers have performed a thorough work and the authors provided clear responses during rebuttal. The only relevant concern has to do with some remaining doubts on inter-annotator agreement. This information should be clearly stated in the paper, together with all other minor comments raising during the discussion period.

---

### Decision · Program_Chairs · 2023-10-07

**Decision:**

Accept-Main

**Comment:**

This work presents a multi-task NLP benchmark and analysis platform for evaluating Swedish language models.

The three reviewers agree on the strong/excellent quality of the work with respect to soundness and excitement. The reviewers have performed a thorough work and the authors provided clear responses during rebuttal. The only relevant concern has to do with some remaining doubts on inter-annotator agreement. This information should be clearly stated in the paper, together with all other minor comments raising during the discussion period.